# Harnessing Artificial Neural Networks for Spinal Cord Injury Prognosis

**DOI:** 10.3390/jcm13154503

**Published:** 2024-08-01

**Authors:** Federica Tamburella, Emanuela Lena, Marta Mascanzoni, Marco Iosa, Giorgio Scivoletto

**Affiliations:** 1Department of Life Sciences, Health and Health Professions, Link Campus University, 00165 Rome, Italy; f.tamburella@unilink.it; 2Spinal Center, Spinal Rehabilitation Laboratory, IRCCS Fondazione S. Lucia, 00179 Rome, Italy; e.lena@hsantalucia.it (E.L.); m.mascanzoni@lumsa.it (M.M.); g.scivoletto@hsantalucia.it (G.S.); 3Department of Psychology, Sapienza University of Rome, 00183 Rome, Italy; 4Smart Lab, IRCCS Fondazione Santa Lucia, 00179 Rome, Italy

**Keywords:** spinal cord injury, outcome, prognosis, artificial neural networks

## Abstract

**Background:** Prediction of neurorehabilitation outcomes after a Spinal Cord Injury (SCI) is crucial for healthcare resource management and improving prognosis and rehabilitation strategies. Artificial neural networks (ANNs) have emerged as a promising alternative to conventional statistical approaches for identifying complex prognostic factors in SCI patients. **Materials:** a database of 1256 SCI patients admitted for rehabilitation was analyzed. Clinical and demographic data and SCI characteristics were used to predict functional outcomes using both ANN and linear regression models. The former was structured with input, hidden, and output layers, while the linear regression identified significant variables affecting outcomes. Both approaches aimed to evaluate and compare their accuracy for rehabilitation outcomes measured by the Spinal Cord Independence Measure (SCIM) score. **Results:** Both ANN and linear regression models identified key predictors of functional outcomes, such as age, injury level, and initial SCIM scores (correlation with actual outcome: R = 0.75 and 0.73, respectively). When also alimented with parameters recorded during hospitalization, the ANN highlighted the importance of these additional factors, like motor completeness and complications during hospitalization, showing an improvement in its accuracy (R = 0.87). **Conclusions:** ANN seemed to be not widely superior to classical statistics in general, but, taking into account complex and non-linear relationships among variables, emphasized the impact of complications during the hospitalization on recovery, particularly respiratory issues, deep vein thrombosis, and urological complications. These results suggested that the management of complications is crucial for improving functional recovery in SCI patients.

## 1. Introduction

Spinal cord injury (SCI) leads to several severe symptoms, including motor deficits, sensory deficits, and autonomic nervous system dysfunction. SCI results in persistent disabilities for patients and has a notable socioeconomic impact. Historically, traumatic spinal cord injuries (tSCI) have been more prevalent among young adult men and women, primarily resulting from car accidents, falls from heights, and sports-related incidents [1]. Previous studies on the age distribution of tSCI showed a primary peak among individuals aged 10–29 years over three decades ago; however, since the 2000s, the age of this peak group has gradually increased. The reported incidence of tSCI varies widely between countries and even between regions within the same country [2]. Recent data indicate that the incidence of tSCI in high-income countries ranges from 12.6 to 86 cases per million [3,4,5,6,7,8].

Understanding the up-to-date epidemiology and demographic characteristics of both tSCI and non-traumatic SCI is crucial for managing healthcare resources, as well as the understanding of the prognosis of neurological and functional outcomes after SCI is essential for responding to patients’ questions regarding their potential functional capabilities. Moreover, it helps in determining the required resources for inpatient rehabilitation and post-discharge care. Additionally, having a thorough understanding of the trajectory and factors influencing the natural recovery of SCI has become a scientific necessity. This understanding is essential for assessing also the effectiveness of new pharmacological and rehabilitative treatments [9].

Outcome prediction of neurorehabilitation is determined by prior knowledge of clinical and demographical factors, including functional and clinical scale scores, level of lesion, presence of comorbidities, as well as age and gender, useful for a rehabilitative prognosis. Classical statistical approaches are based on linear or binary regressions, taking into account many different variables assessed at admission into the rehabilitation hospital and, in some cases, variables assessed during the hospitalization or at discharge. Over the past few years, many different prognostic factors have been identified.

In tSCI, several features such as age, the initial International Standards for Neurological Classification of Spinal Cord Injury (ISNCSCI) evaluation [10], the Magnetic Resonance Imaging (MRI) characteristics of the lesion, the presence of comorbidity and the occurrence of complications (in particular pneumonia) [11] have been correlated with the neurological and functional recovery. Consequently, interventions aimed at mitigating cognitive changes, post-injury pneumonia, and unhealthy body weight can enhance neurological improvement in SCI subjects. Regarding ambulation prognosis, a 2011 study introduced a straightforward and precise prediction rule for independent ambulation outcomes after tSCI considering age, motor scores of the myotomes L3 and S1, and light touch sensation of dermatomes L3 and S1 [12].

However, the relationships between different factors (such as demographical, clinical, biological, and psychological ones) and outcomes could not be linear and interlaced. Due to the above-mentioned, it is possible that most traditional approaches may not reveal such complexity [13,14]. Artificial Neural Networks (ANNs) are recently emerging as an alternative approach that is more accurate for discriminating between different categories of possible prognostic factors useful for predicting outcomes in patients needing neurorehabilitation, such as people who had a stroke or a traumatic brain injury [13]. Outcome prediction in neurorehabilitation is basically determined by prior knowledge of demographical variables, physical factors, and the presence of comorbidities to evaluate for clinical and rehabilitative prognosis [15]. Conventional statistics identified the variables directly associated with the outcome (eliminating the variables correlated to other variables correlated to the outcome), able to explain the variability among patients. The new approach allowed by Artificial Neural Network had the advantages of assigning a weight to each variable taken into account, having the possibility to consider complex relationships between input and output, with models extracted by data and not prior hypothesized by researchers, increasing the accuracy of predictions [13]. The types of ANN used in rehabilitation can be divided according to supervised and unsupervised approaches, further divided into classification and regression or clustering and dimensionality reduction, respectively. The most common algorithms for machine learning used in ANNs are support vector machine, k-nearest neighbors, naïve Bayes, decision tree, and others, with an accuracy usually ranging between 78 and 98% [13].

For patients with spinal cord injury, a recent systematic review of the literature highlighted that the use of artificial neural networks able to perform machine learning has good potential in diagnosis, prognostication, management, rehabilitation planning, and risk prevention of chronic complications and mental illness [16]. However, the same review identified two main problems in the studies concerning the use of ANNs for identifying the prognostic factors in individuals with SCI. The first one is the need to validate the ANN-based predictive scoring systems with respect to other approaches and with respect to the used algorithms that include the selection of the input variables. The other problem highlighted in that review was that most of the analyzed studies investigated many input variables but with small datasets, going from only 9 patients up to a study with 862 patients (in mean: 165 patients for each study) [16]. Another problem concerning the use of ANN as a prognostic classifier is its reliability. In fact, the ANN could achieve the same level of prognostic accuracy, assigning different weights to the synapses and, hence, suggesting different levels of importance for the input variables as prognostic factors [17].

Therefore, this study has two aims: (1) to evaluate the clinical and demographic factors affecting the functional outcome of a large population with SCI by means of ANN, and (2) to compare the prognostic value of ANN with that or a more conventional statistical approach, and deeply analyzing also the problems of ANN for understanding the problems and identifying solutions.

## 2. Materials and Methods

### 2.1. Patients and Data

This study was a secondary analysis conducted on a large database of 1308 patients already used in some previous studies [18] and furtherly alimented with new data. A retrospective analysis was performed on the medical documentation of individuals who suffered from either traumatic or non-traumatic SCI, regardless of the level or the severity of the injury, age above 18 years, and who underwent the first admission for neurorehabilitation after the lesion. We excluded the medical records of the patients with cognitive impairments, which prevent a full neurorehabilitation program or a proper evaluation. Furthermore, being the functional status at discharge the focus of the present study, we also excluded patients who were discharged or transferred for over 3 weeks and then readmitted, the subsequent readmissions categorized as secondary admissions, and patients with admission durations of less than 7 days. For the same reason, we excluded patients who died during their rehabilitation stay. These patients were admitted to our spinal cord unit for rehabilitation treatment following the injury, spanning from 1996 to 2023.

Because our hospital is also a research institute, at admission, all patients signed an informed consent for the utilization of their data in translational research.

### 2.2. Clinical Assessment and Neurorehabilitation

Our neurorehabilitation ward in Fondazione Santa Lucia is intended for subacute rehabilitation. It is composed of a wide gym with devoted areas for individual treatments, and each bedroom has two beds. For each patient with SCI, the neurorehabilitation program was planned by a pool of neurologists and physiatrists. Neurorehabilitation is conventionally administered by therapists 6 days a week, at least 2 sessions per day, each session lasting 40 min. In line with patients’ needs, individual therapy could include other types of neurorehabilitation besides physical therapy, such as cognitive, occupational, or hydrotherapy rehabilitation, as well as specific therapy for breathing, swallowing, bowel, and bladder dysfunctions. All rehabilitation treatments began within 24 h from admission.

At admission, the database was filled with the following: (i) 3 continuous variables: age, Spinal Cord Independence Measure (SCIM) version II [19] or III-score [20], Walking Index for Spinal Cord Injury (WISCI)-score [21]; (ii) 2 ordinal variables: the American Spinal Injury Association Impairment Scale (AIS) score (A, B, C or D) [10], lesion level (cervical, thoracic or lumbar) and 11 binary variables: gender, etiology (1:traumatic or 0:not), surgical intervention (1:yes or 0:not), presence of pressure sores (1:yes or 0:not), heterotopic ossifications (HOs) (1:yes or 0:not), respiratory complications (1:yes or 0:not), pulmonary embolism (1:yes or 0:not), deep vein thrombosis (1:yes or 0:not), urologic complications (apart from urinary infections) (1:yes or 0:not), preservation of motor tracts (1:motor completeness or 0:not), other complications (1:yes or 0:not).

During the hospitalization (the mean length of stay in the hospital was 140 ± 106 days), the following 8 binary parameters (absent/present), already collected at admission, were recorded into the database: occurrence of generic complications, pressure sores, HO, respiratory complication, pulmonary embolism, deep vein thrombosis, urologic complications, other types of complications. Many different variables were routinely recorded also at discharge, but for this study, only the SCIM-score was analyzed as the primary outcome of neurorehabilitation.

### 2.3. Artificial Neural Network

ARIANNA model (ARtificial Intelligent Assistant for Neural Network Analysis) was used to conduct the Artificial Neural Network analysis. It was already used in previous studies for identifying the prognostic factors [13,17,22,23]. This model is a Multilayer Perceptron Procedure formed by the input layer (from which entered the above-listed variables), two hidden layers (of 5 elements each), and a final output layer (the predictor of the dependent variable). The architecture of the ANN was that of a Feed Forward Neural Network (FFNN), with data moving in only one direction, from the input nodes through the two hidden layers to the output node. The activation function for all the units in the hidden layers and for the output layer was a hyperbolic tangent. The chosen computational procedure was based on online training. The ANN was developed in the Statistical Package for the Social Sciences software (SPSS) of IBM, version 23. Figure 1 shows a schematic representation of ARIANNA applied to this study in which 16 input variables were initially considered. Disadvantages of using ANN include its black box nature, greater computational burden, proneness to overfitting, and the empirical nature of model development [24]. In particular, the capacity to overfit the data implies that the ANN could achieve the same level of performance with different weight assignations, a characteristic that could allow the achievement of high accuracy but reduce reliability [17]. To compensate for the poor reliability of ANN, for each analysis, ARIANNA ran 10 times, and the predictions more correlated with the actual outcome were selected.

The ANN also provided the Raw Indices representing the percentage weight associated with each input variable (the sum of all Raw Indices is 100%). The Normalized Indices were computed as the percentage of the highest weight (the Normalized Indices = 100% is assigned to the variable having the highest RI). These values provided information about the prognostic power of each assessed variable.

### 2.4. Statistical Analysis

Both continuous and ordinal variables were reported in terms of mean and standard deviation, whereas binary and nominal variables were pointed out as percentage frequencies. Predictive scores were selected for reporting the ANN results, raw percentage importance (RI) of each variable (the total of all RIs is 100%), and normalized importance (NI, with respect to the most important variable). All data were used for training the ANN and not split for testing. The hyperparameters were set as follows. The training criteria was an online batch, the optimization performed by scaled conjugate, and the initial values were lambda = 0.0000005, sigma = 0.00005, interval center = 0, interval offset = 0.5 mem size = 1000, the error steps = 1 (data = auto) training timer was on with a max time = 15 and automatic max epochs. Error change was = 1.0 × 10^−4^, and the error ratio = 0.001. The mean absolute error (MAE) was used to assess the performance of ANN

A linear regression analysis was also conducted to identify, among the analyzed factors, those significantly entering into the model. The linear regression had the same input variables of ANN, and the same predicted output that was the SCIM-score at discharge. Particularly as regards the SCIM scale, the SCIM score assessed at admission to the neurorehabilitation hospital was one of the input variables of both the ANN and regression model. Furthermore, the predicted variable for both models was the SCIM score assessed at discharge. Unstandardized (B) and standardized (Beta) coefficients of linear regressions were reported together with *p*-values of variables entered (*p* ≤ 0.05) or not (*p* > 0.05) into the model. Pearson coefficient was used to assess correlations, and linear fitting of prediction versus actual outcomes was performed using the least squares method. Mean absolute error (MAE) was used to compare the absolute differences between computed and actual scores. For the statistical analyses, as well as the ANN, SPSS (Statistical Package for the Social Sciences) software of IBM, version 23, was adopted

## 3. Results

A database of 1308 patients with spinal cord injury was analyzed. Of these 1308 patients with SCIM assessment performed on admission to the neurorehabilitation department, 18 were transferred for emergency, and 34 died during hospitalization. Of the remaining 1256, 305 of them did not have complete data and were therefore excluded from the statistical analysis, which was then conducted on a sample of 951 patients (see Figure 2).

The average days elapsed between SCI and hospitalization was 51.6 ± 64.1. The sample had a mean age of 51.5 ± 18.4 years and was formed by 69% of males. A traumatic etiology of spinal cord injury was recorded in 43% of the cases. The most common injury level was the thoracic one (47% of the individuals), and the SCI was mainly a sensory-motor incomplete one (AIS C: 27%; AIS D: 35%). Sixty-six percent of patients underwent a surgical intervention before neurorehabilitation. At admission, 84% of patients included were unable to ambulate (WISCI score equal to 0), while only 2.8% were autonomous in ambulation (WISCI score equal to 20). The remaining patients could ambulate but with different levels of physical assistance, devices, or orthosis [21]. The mean SCIM score at admission was 26.2 ± 20.6, whereas at discharge, it was 61.3 ± 26.3. Other data are reported in Table 1.

In Table 1 we also reported the raw and NI of each prognostic factor as assessed by the ANN (on 951 patients having complete data). The reliability of ANN throughout the 10 runs was measured by the standard deviation of RI coefficients, which ranged between 0.7% for surgical intervention and 2.5% for motor completeness and SCIM, and by the Cronbach’s alpha computed on RI values that were 0.987. The linear regression identified seven statistically significant variables that entered into the model, and also these results are reported in Table 1.

As shown in Table 1, the two approaches had in common the identification of the SCIM at admission as the most important predictor, followed by age, level of lesion, ASIA score, and presence of pressure sores at admission. A role was also played by traumatic etiology in both models. The ANN associated a high importance also to WISCI score at admission and to motor completeness of the lesion, complications and presence of deep vein thrombosis at admission. Linear regression also identified a significant role for gender, which was marginal for ANN.

The correlation between the SCIM scores at discharge and those predicted by ANN was R = 0.75, just slightly superior to that with values predicted by classical regression R = 0.73 (both were highly statistically significant, *p* < 0.001). The MAE with respect to the SCIM-score at discharge was 13 ± 11 points for the values predicted by ANN and 15 ± 10 for linear regression.

Figure 3 shows the predictions of ANN versus the actual SCIM score assessed at discharge. The linear regression had the disadvantage of also predicting values out of the range of the SCIM (>100). Adjusting values for covering this range did not improve the performance of the linear regression predictions (R = 0.73, MAE: 15 ± 10). The ANN did not provide values out of the range, but it was evident that it overestimated the low score of SCIM, corresponding to the poor outcomes.

To test if this overestimation was related to complications that occurred during the recovery into the hospital, we have performed a new analysis with the ANN, including eight other variables (i.e., surgical intervention, presence of pressure sores, HOs, respiratory complications, pulmonary embolism, deep vein thrombosis, urologic complications or other complications) recorded during the recovery. The computed values are shown in Figure 3 with grey dots. The correlation increased to R = 0.87, and the MAE was reduced to 10 ± 8. The most important prognostic factors are reported in Table 2.

The importance of variables at admission was reduced by 24.6%, covered by the variables assessed during hospitalization. In particular, pulmonary embolism, deep vein thrombosis, or urologic complications occurred during hospitalization, weighted for 14.4% of the prediction.

## 4. Discussion

We investigated the feasibility of predicting the activity of daily life competency (SCIM Score) at discharge from rehabilitation, from demographic and clinical features at admission using data extracted from a single center, a large database of individuals with traumatic and non-traumatic SCI who underwent inpatient treatment at a spinal center in Italy. To perform this analysis we used an ANN (ARIANNA), but we also performed a linear regression analysis to compare the two methodologies.

For both approaches, the level of independence (SCIM score) achieved at discharge from rehabilitation strictly depended on demographic (mainly age) and clinical (injury level, ASIA score, presence of complications, in particular pressure ulcers, etiology of injury) variables. The ANN analysis also highlighted the significance of the WISCI score, motor completeness of the lesion, and, although marginal, a gender influence.

Findings about demographic and clinical variables corroborate existing literature. Starting from the former, research on the impact of age on SCI rehabilitation outcomes suggests that younger individuals generally achieve better recovery compared to older adults. This higher recovery in younger patients is often attributed to greater neural plasticity and a lower incidence of comorbid conditions, which enhance their ability to adapt to injury [25]. Conversely, older adults with SCI face challenges due to age-related factors such as reduced muscle mass, decreased bone density, and the presence of comorbidities, all of which can impede the rehabilitation process and restrict functional recovery. Additionally, the slower neurological regeneration in older patients can limit the degree of possible improvement through rehabilitation efforts [26]. Age data are intriguing, considering that, recently, SCIs have seen a rise in cervical hyperextension injuries due to falls on level surfaces and low falls among older people [26]. The World Health Organization reported that the worldwide population enhanced from 3.6 billion in 1970 to 5.3 billion in 1990 and exploded to 8.0 billion by 2022. It is estimated that by 2050, the world’s population aged 60 years and older will reach over 2.0 billion [27], with the expectation that the mean age at the time of injury will rise as the population ages [28]. Furthermore, the heightened risk of reoccurring falls and subsequent fractures can enhance the risk of mortality for individuals sustaining fall-related SCIs [29]. Falls are considered a category of tSCI, as well as violence and car/motorbike accidents with high hazards for mortality [29]. Compared to tSCI, non-traumatic SCI patients were significantly older, more numerous, and mainly affected by incomplete lesions and associated with paraplegia. Despite no significant differences were found in the length of stay, tSCI patients showed greater improvement when discharged [30]. Our data indicate an association between the etiology of the injury and the level of independence, although weaker than the other relationships identified.

Regarding the other factors, the ANN identifies gender as a marginal prognostic factor compared to others. Gender was statistically significant as a prognostic factor, but it has the smallest weight among those identified. This finding could align with the conflicting data present in the literature, related to a small, sometimes significant, effect. To date, there is no universal consensus on the gender-prognosis correlation for recovery. Authors previously demonstrated on 281 patients that gender does not seem to influence rehabilitation outcomes even if male and female individuals showed significant epidemiological differences. Women with SCI had a lower frequency of traumatic lesions and complications at admission associated with a higher frequency of incomplete injury (ASIA impairment C) [31]. On the other hand, research suggested that the SCI severity, as well as the ultimate recovery of motor function after the lesion, is significantly influenced by gender. In fact, motor function recovery is remarkably higher in females, possibly due to the effects of estrogen on pathophysiological processes as gender-specific mechanism(s) of neuroprotection, although not yet elucidated [32].

The observed correlations between higher SCIM scores at discharge and lesions at the lumbar or thoracic spinal cord (i.e., less severe lesions) and younger age are consistent with already published data. Both the level and severity of spinal cord injury (SCI) greatly influence rehabilitation outcomes, with literature emphasizing the importance of the anatomical location of the injury in determining functional recovery and quality of life [33]. Individuals with cervical SCI typically face greater rehabilitation challenges than those with thoracic or lumbar injuries, largely because of the more significant effects on upper limb function and overall mobility [34]. Furthermore, the severity of the lesion scored on the ASIA impairment scale correlates with the degree of functional impairment and the likelihood of meaningful recovery [35]. The individuals classified as AIS A, B, and C show better improvements in functional independence if the SCI is localized at a caudal level compared to those with a cervical injury. Similarly, individuals with complete SCI (AIS A) have higher scores in the SCIM if they have an injury at a caudal level [35]. The SCIM is a functional outcome measure worldwide adopted because it is specifically designed to assess the level of independence of subjects with SCI. The SCIM evaluates the autonomy of patients in different areas: self-care (score 0–20), respiration and sphincter management (score 0–40), and mobility (score 0–40) for a total score ranging from 0 (lowest level of independence) to 100 (complete independence). The SCIM is frequently adopted as part of SCI clinical trials and observational cohorts, and it is often used because bowel and bladder functions, endpoints commonly understudied, are part of the SCIM assessment. Based on the findings of a systematic review by Anderson et al. [36], the SCIM has been found to be the most valid and useful measure of functional recovery in the SCI population for clinical and research purposes, suggesting it is the most sensitive, reliable, and valid measure of SCI-specific global disability.

Furthermore, recently, it has been reported that the rehabilitative goals selected by the Physical Therapist in SCI management during recovery are strictly influenced by the AIS score and the lesion level [37]. Consequently, it is crucial in the neurorehabilitation pathway to tailor ad hoc rehabilitation strategies considering both lesion level and severity, as well as the greater difficulties of an older patient, to better enhance outcomes and independence in the Activities of Daily Living. Besides the significant differences in rehabilitation needs and recovery potential among individuals with cervical, thoracic, and lumbar injuries, it should be considered that non-traumatic lesions could have minor benefits after rehabilitation therapy in comparison to traumatic SCI. This difference may be related to an often insidious presentation and a slow timing of diagnosis at the admission to a rehabilitation unit. According to this evidence, it is crucial to plan and manage various neurorehabilitation interventions in the different categories of SCI. Consequently, to obtain a better functional recovery for these patients, it is necessary to elaborate specific rehabilitation programs for more compromised individuals (e.g., older subjects, non-traumatic SCI, etc.).

With regard to the impact of complications at admission and during hospitalization, mainly the pressure sores, our study is in agreement with previous ones demonstrating that the presence of complications has a negative impact on patients’ functional status at discharge [38,39,40]. To gain a deeper understanding of this relationship, various factors must be considered, particularly the concept of “immobility” required for healing complications. This is especially pertinent for pressure ulcers, as this approach may delay attaining a sitting position, utilizing a wheelchair, and ultimately achieving comprehensive rehabilitation goals [41]. It is also important to consider that complications, particularly pressure ulcers, induce a chronic inflammatory state characterized by anemia, low serum iron, hypoproteinemia, and hypoalbuminemia [42]. These conditions can significantly diminish the functional potential of patients [43]. Irrespective of the previously mentioned factors influencing recovery. Furthermore patients experiencing complications tend to have extended hospital stays, receiving fewer hours of rehabilitation or less intensive rehabilitation compared to individuals without complications [41].

The second aim of the study was to compare the prognostic value of ANN with that of a more conventional statistical approach, considering that ANNs may manage large databases and provide accurate predictions of outcomes of neurorehabilitation. Previous studies on stroke disease showed that their accuracy was higher than that of conventional statistical approaches such as linear regression [22,23]. However, this was not always true, especially for the artificial neural networks developed many years ago [44] or when the results of ANN have been compared to specific statistical methods such as cluster analysis [22].

Attempts to use ANN for predicting the prognosis of patients suffering a SCI were recently made. In the review by Håkansson et al. [45], it was reported that the most commonly used outcomes in the few studies present in literature were AIS grades, walking ability, or SCIM scale. Kishikawa et al. [46] were able to predict the prognosis of patients with cervical SCI from acute clinical data using ANNs, for which prediction results were based on the ASIA Impairment Scale. Even if the Authors declared that the AIS score alone can only provide a general understanding of the condition of patients with cervical SCI, ANNs developed from the data of the acute phase predicted the prognosis of the patients with cervical SCI more accurately than multiple regression analysis. Draganich et al. [47] suggested that ANN method is feasible for classifying outcomes in SCI and may provide improved accuracy, sensitivity and specificity than multiple linear regression analysis in identifying which individuals are less likely to ambulate and may benefit from specific rehabilitative strategies. Belliveau et al. predicted the prognosis after tSCI using the ANN [48] and compared the results with a multiple linear regression analysis setting. They set as the main outcome the self-reported ambulation and reported a high level of predictive accuracy with ANN, and they also predicted independence in self-care activities with moderate accuracy. A recent study conducted on 210 patients with spinal cord injury compared different ANNs for predicting the SCIM-score at outcome [14]. The MAE was in a mean of 8.6, slightly lower than that of our study (that was an average of 13 points). Despite the authors performing a linear regression on their data, they did not compare the accuracy of the two approaches in that study. Dietz et al. reported as the main problems of ANN-based predictive scoring systems the necessity of validating this approach with respect to other approaches, the small datasets in a mean of 165 patients for each study, and the reliability of using ANN as a prognostic classifier [16]. In our study, we analyzed the data by overcoming these problems with a sample of 1256 patients, a comparison with the conventional statistical regression analysis method and by analyzing the reliability measured by the standard deviation of RI coefficients. Our analysis, performed on an extensive dataset of 951 individuals with SCI, underpins that for both approaches, the R value exceeds 0.70, indicating that they can be considered reliable predictors of the SCIM-score at discharge. Additionally, the nuanced higher R-value obtained with ANN suggests that this new approach is not less valid than linear regression, but slightly superior to linear regression in predicting SCIM output.

Our first analysis showed a correlation between predicted values and actual outcomes of 0.75. This analysis identified the variables with the higher weight on the outcome and can provide clinicians the possibility to provide a real prediction of the outcome already at the admission. Our second analysis tried to identify the complications that occurred during the hospitalization that affected the capacity of ANN to predict the SCIM score at discharge. Taking into account these further variables recorded during the hospitalization, the correlation improved at 0.87, providing important information about the weight of these adjunctive variables. However, this approach did not provide a real possibility to predict the outcome because the complications occurred during the recovery. This approach is not uncommon and is also used in classical regression analyses, for example, when the length of stay [49], transfer to other hospitals [50], or medical complications [51] (all data not known at admission) are used as independent variables for predicting the rehabilitation outcome. On the other hand, our second analysis underlined which complications may greatly affect the outcome, suggesting to clinicians to importance of allocating the resources to avoid them. From a clinical management point of view, this approach can be defined as having “Continuous Valued Input”, and the machine learning is iterated with additional variables [52].

In terms of prognostic factors, both approaches agree on the importance of age and some clinical variables discussed above. The main discrepancies between the two approaches, were related to the WISCI score and motor completeness that were not taken into account by the linear regression, whereas weighted for 21% on the ANN prediction. Conceivably, it was due to the fact that linear regression aims to find the smallest number of possible predictors, and these two factors are strictly connected with the SCIM score at admission. In fact, the linear regression identified only seven statistically significant variables that entered into the model, whereas ANN took into account all the input variables. It means that covarying variables can be associated with a high level of prognostic weight by the ANN, but not by the linear regression.

ANN was more complicated than linear regression, and the small improvements in its performance seemed to not justify its use. This improvement is small because the ANN tended to overestimate the patients who had a poor outcome. As shown in our secondary analysis, it could be explained by the fact that these subjects often had complications during the hospitalization. These complications accounted for about one-fourth of the outcome, and they were difficult to predict at admission to the neurorehabilitation hospital. Another notable element is that the ANN analysis highlights the prognostic role of complications, both those present at admission (see Table 1) and those that arise during hospitalization (see Table 2). Among the latter, the most important is the possibility of developing pulmonary embolism, deep vein thrombosis, or urological complications (NI score ranging from 24.9 to 18.2).

This is very intriguing considering that a recent study [53] highlighted that since 2010, the highest mortality rate after SCI is for respiratory diseases and that the overall age-standardized mortality rate was three times higher for individuals with SCI than the general population. This study also stressed that a key element for improving life expectancy after SCI is to reduce mortality rates from respiratory diseases. In line with these data, in our study, the ANN identifies respiratory issues that occurred during hospitalization as the most important complication affecting recovery. This element, combined with the literature data, indicates that special attention should be paid not only to the characteristics of SCI mentioned above but also to the respiratory complications that occur during hospitalization. This connection is even more evident when considering the strong relationship between respiratory complications and deep vein thrombosis. It is worth noting that venous thrombosis is associated with considerable short-term morbidity and mortality. The mortality rate after venous thrombosis is about 20% within one year, but this data is 2 to 4 times higher for patients with pulmonary embolism. Furthermore, 10–20% of these patients die within three months after the event [54]. The third type of complication to consider for the prognosis that the ANN has highlighted is the urological complications. Recurrent urinary tract infections are mainly developed in most of all SCI patients, with an incidence peaked in the 1st and 10th 5-year intervals. Furthermore, the most common complications were bladder stones, hydronephrosis, and vesicoureteral reflux. In the case of male gender, cervical SCI, and condom catheter use, the risk of urologic complications is higher, but almost all patients with SCI are at risk of developing urinary complications. In fact, also for patients who are supposed to be “stable” from the urological point of view, it is necessary to underwent regular follow-up and surveillance [45].

The main use of artificial intelligence in predicting outcomes seems to be not simply the replication of classical statistical approaches with a small improvement in terms of accuracy against a complication of the model and a reduction of reliability. Artificial intelligence may instead provide continuous monitoring and computation of the clinical parameters of the patient, identifying the potential role of complications in real-time on the outcome. Cerasa and colleagues [13] already claimed that, in front of good performance of specific machine learning algorithms (such as ANN), there is high heterogeneity in features extracted from low-dimensional clinical datasets that may reduce the enthusiasm for applying this powerful method in clinical practice. They also claimed the need to better capture and predict the dynamic changes in patients. Our findings about the role of hospitalization complications support this point of view. Dietz and colleagues [16] also suggested a potential role of machine learning approaches for personalizing rehabilitative care.

Furthermore, there is a growing body of literature about the use of ANN and Machine Learning techniques to predict the rehabilitation outcomes in different neurological pathologies. The first models have been proposed by researchers who home-made programmed these tools (for a review Cerasa et al. [13]). Nowadays, statistical software includes easy-to-use tools for setting the parameters of artificial neural networks. Also, ARIANNA, the ANN used in this study, is based on a set of hyperparameters previously determined [22] and used in the ANN tool of SPSS. In the near future, as well as for image or text Artificial Intelligence (AI) generators, it will be possible to just ask the AI for the required model, and the AI will program it or use it. At the moment, Chat-GPT version 4 (OpenAI) is already able to perform a prediction of rehabilitation outcomes using online models if the operator provides the required data to give as input. This possibility opens a fascinating scenario in which clinical professionals can easily use AI, but they should be trained about the pitfalls and drawbacks of AI.

Given that ANN can be modified by incorporating other clinical parameters, even if adding parameters to ANN does not necessarily improve accuracy, ANNs have much potential to be upregulated with new effective variables in the future. Future studies should investigate other important outcomes, such as the length of stay, the days between SCI and admission into the neurorehabilitation ward, the mortality, and the destination after discharge from neurorehabilitation hospitals. Knowledge of even these elements in relation to patient prognosis will allow clinicians to perform effective rehabilitation based on ANN data, possibly leading to the improvement of the patient’s quality of life after discharge. In addition to the lack of integration of data such as mortality or the others mentioned above, this study has other limitations. Data were collected for SCI patients hospitalized in a highly specialized center of neurorehabilitation. Therefore, it is possible that the current predictive model will not be able to accurately predict the prognosis of patients in less specialized facilities or facilities with less time for neurorehabilitation. To manage this issue, when predicting prognosis at other hospitals, it could be possible to incorporate rehabilitation time and type as clinical parameters. In the future, we should consider ways to utilize different predictive models, but further research is required for this purpose. Furthermore, the ANNs are black boxes concerning the inter-relationships among variables. For this reason, it is possible that covarying variables received similar weights (such as SCIM at admission, lesion level, and ASIA grade, all strictly each other interconnected), whereas conventional regressions, analyzing the residuals from the second step, minimize the overlap of variables quantifying similar information

## 5. Conclusions

This study assessed the feasibility of predicting daily living ability (SCIM Score) at discharge from rehabilitation using demographic and clinical data from a large single-center database of individuals with traumatic and non-traumatic SCI. Both an ANN and linear regression were used for analysis. Key findings indicate that SCIM scores at discharge are significantly influenced by age, injury level, ASIA score, presence of complications (especially pressure ulcers), and injury etiology. The ANN also highlighted the importance of the WISCI score, motor completeness, and a marginal gender influence. Younger patients generally achieve better recovery due to greater neural plasticity and fewer comorbid conditions, while older adults face challenges such as reduced muscle mass and slower neurological regeneration. Traumatic SCI patients show greater improvement compared to non-traumatic SCI patients despite similar lengths of stay.

The amount of clinical and administrative data that healthcare providers generate has seen explosive growth over the last few years and it could be usefully utilized by ANN for predicting the outcomes of a patient and also for continuous monitoring of the clinical path. Our study showed that ANN provided a small advantage with respect to classical statistic methodologies, but it will be possible to implement a continuous updating of ANN predictions with new data recorded during rehabilitation. In fact, ANN increased its accuracy when fed with data recorded during the hospitalization and not only with data recorded at admission. ANN emphasized the impact of complications during hospitalization, particularly respiratory issues, deep vein thrombosis, and urological complications, on recovery, suggesting that the management of complications is crucial for improving functional recovery in SCI patients. Future research should explore other important outcomes, such as length of stay and discharge destinations, to refine these prognostic models further.

## Figures and Tables

**Figure 1 jcm-13-04503-f001:**
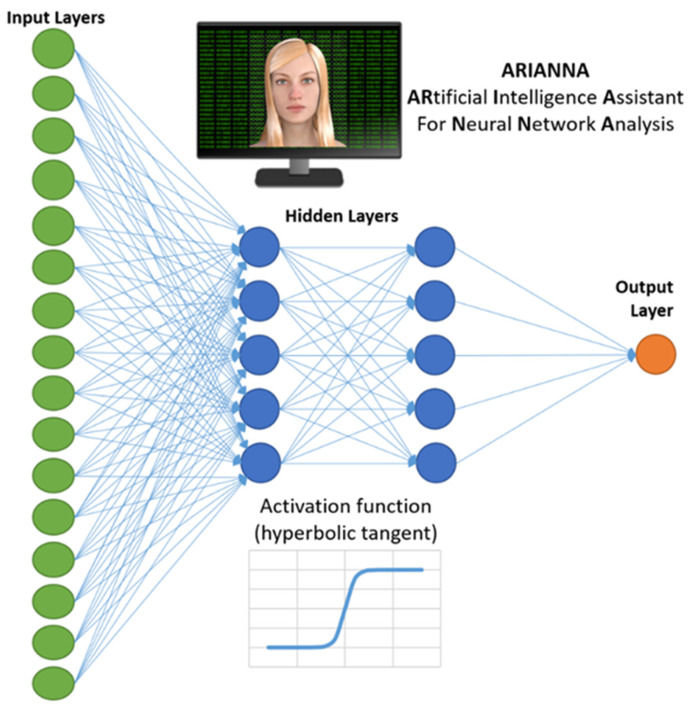
ARIANNA, ARtificial Intelligent Assistant for Neural Network Analysis (16 input layers, 2 levels of 5 hidden layers, 1 output layer).

**Figure 2 jcm-13-04503-f002:**
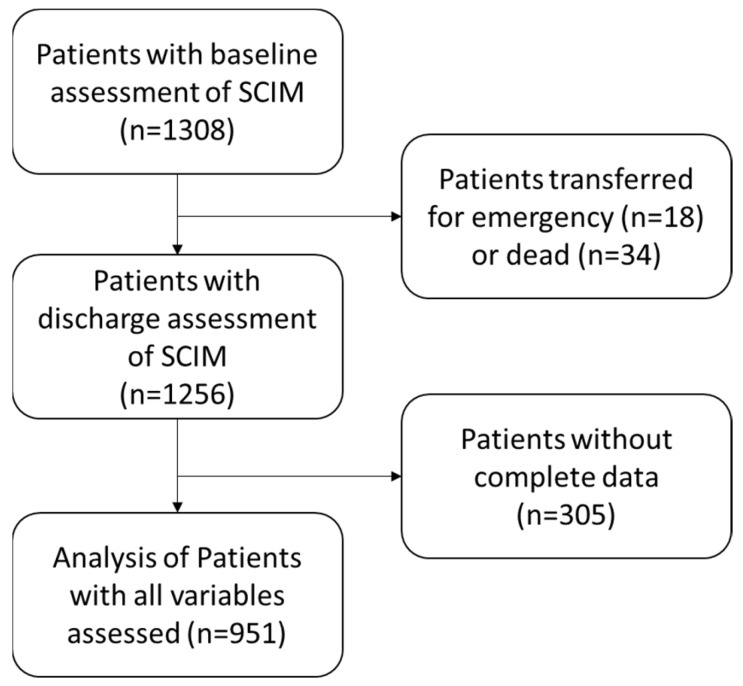
Flow chart of the data analysis.

**Figure 3 jcm-13-04503-f003:**
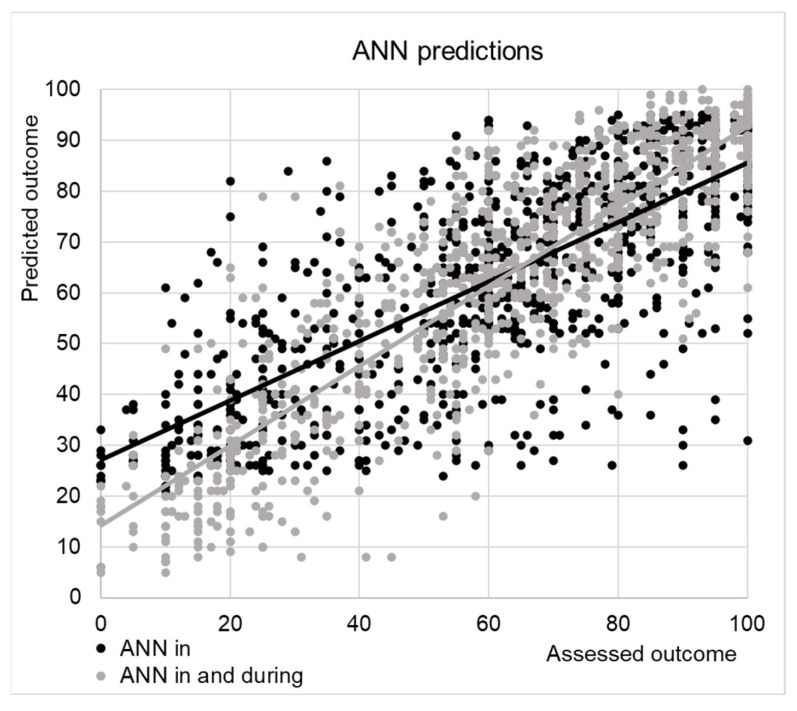
SCIM-scores predicted by ANN with 16 input layers related to the admission variables (black dots) and with 24 input layers, also including eight variables recorded during the hospitalization. Relevant linear regression lines are also shown.

**Table 1 jcm-13-04503-t001:** Parameters assessed at admission, their mean ± standard values (for continuous and ordinal variables) or percentages of occurrence (for binary variables), the raw (RI) and normalized importance (NI) identified by ARIANNA, the unstandardized (B) and standardized (Beta) coefficients of linear regressions followed by the *p*-values of being or not into the model. Variables were ordered according to their importance in the ANN.

Variables at Admission		Artificial Neural Network Analysis		Linear Regression		
Parameters	Descriptive Statistics	RI	NI	B	Beta	*p* Values
SCIM	26.2 ± 20.6	19.9%	100.0%	0.517	0.409	**0.000**
WISCI	2.0 ± 5.2	15.2%	76.5%			0.609
Age (years)	51.5 ± 18.4	12.6%	63.2%	−0.438	−0.308	**0.000**
Lesion level	C: 34%, T: 47% L:19%	10.3%	51.5%	−5.740	−0.159	**0.000**
ASIA score	A:29% B:9% C:27% D:35%	6.6%	33.3%	−5.931	−0.268	**0.000**
Motor completeness	35.9%	5.8%	28.9%			0.964
Pressure sores	25.5%	4.5%	22.8%	−9.300	−0.151	**0.000**
Complications	28.3%	4.3%	21.6%			0.407
Deep vein thrombosis	0.5%	4.2%	21.2%			0.270
Aetiology (traumatic)	43.2%	4.1%	20.8%	3.109	0.059	**0.024**
Pulmonary embolism	1.0%	3.9%	19.6%			0.564
Having undergone surgical intervention	66.2%	2.9%	14.3%			0.251
Hectopic ossification	1.5%	2.1%	10.4%			0.857
Respiratory complications	2.2%	1.7%	8.5%			0.234
Gender	M:69% F:31%	1.3%	6.4%	−4.715	−0.081	**0.000**
Urological Complications	0.4%	0.6%	3.0%			0.196

*p* values in bold: statistically significant.

**Table 2 jcm-13-04503-t002:** Variables ordered for the raw (RI) and normalized importance (NI). The difference reported in the last column refers to the RI reported here and that reported in Table 1.

Variables	Assessment Time	RI	NI	RI Difference
SCIM	Admission	17.4%	100.0%	−2.5%
Age	Admission	12.1%	69.6%	−0.5%
WISCI	Admission	10.1%	58.1%	−5.1%
Pulmonary embolism	Hospitalization	7.1%	40.5%	-
Pressure sore	Admission	5.9%	33.8%	1.4%
ASIA score	Admission	5.3%	30.3%	−1.3%
Pulmonary embolism	Admission	4.3%	24.9%	0.4%
Lesion level	Admission	4.2%	24.4%	−6.1%
Deep vein thrombosis	Hospitalization	4.1%	23.7%	-
Urological complications	Hospitalization	3.2%	18.2%	-
Complications	Admission	3.1%	17.5%	−1.2%
Gender	Admission	2.6%	14.9%	1.3%
Pressure sores	Hospitalization	2.6%	14.7%	-
Complications	Hospitalization	2.5%	14.2%	-
Respiratory complications	Hospitalization	2.4%	14.0%	-
Aetiology (traumatic)	Admission	2.4%	13.7%	−1.7%
Having undergone surgical intervention	Admission	2.2%	12.5%	−0.7%
Heterotopic ossifications	Hospitalization	2.1%	12.1%	-
Urological complications	Admission	1.8%	10.5%	1.2%
Respiratory complications	Admission	1.6%	9.3%	−0.1%
Motor completeness	Admission	1.3%	7.4%	−4.5%
Deep vein thrombosis	Admission	0.8%	4.9%	−3.4%
Other complications	Hospitalization	0.4%	2.4%	-
Heterotopic ossifications	Admission	0.3%	2.0%	−1.8%

## Data Availability

Data are available upon reasonable request to the corresponding author.

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
