# Peer review of "Harnessing Artificial Neural Networks for Spinal Cord Injury Prognosis"

_jcm, 2024, doi:10.3390/jcm13154503_

Round 1

Reviewer 1 Report

Comments and Suggestions for Authors

The authors predicted the activities of daily living ability (SCIM score) at discharge for spinal cord injury patients using demographic and clinical characteristics at admission to a rehabilitation hospital. This study is a highly valuable as it demonstrates the potential to predict the SCIM score, an important continuous variable regarding the prognosis of spinal cord injury patients.

However, there are several points regarding the variables used and the research design that need to be revised, as outlined below.

1, All patients included in this study were from a rehabilitation hospital, and the number of days from injury to admission was not standardized. The number of days from injury to admission significantly impacts the AIS score and SCIM score, making this variable an important factor. The authors should either incorporate the number of days from injury to admission into the predictive model or specify the average number of days from injury to admission. If this is not possible, it should be noted as a limitation.

2, This study includes complications during hospitalization in the prediction model, making it an important factor. However, if prediction at admission is important, then inclusion of complications during hospitalization is inappropriate. Including these variables may improve prediction accuracy, but given the true value of the prediction model, complications during hospitalization should not be included. Furthermore, it should be added that there is no clear statement regarding the timing of the predictions. If complications are to be included as a variable, the timing of the secondary prediction should be clearly stated and only complications that have occurred by that time should be included. For example, a secondary prediction one week or one month after hospitalization would make the prediction model more practical and realistic.

3, Recent literature on prognosis prediction for spinal cord injury using artificial intelligence is prevalent, but the crucial aspect is its versatility. The discussion should include whether this predictive model can be used by those with little knowledge of artificial intelligence, addressing its versatility and future development potential. For example, the authors should consider whether this predictive model could eventually be developed into a web application. While it is not necessary to detail specific methods, discussing the future possibilities of the model would enhance the comprehensibility of this research's value. 

Reviewer 2 Report

Comments and Suggestions for Authors

This is a well-written study on the use of artificial neural network to predict improvements in patients with spinal cord injury. The study has a high clinical significance, and shows that ANN performs better than traditional linear regression approach, and improves with additional data reflecting the patient's progression. There are several minor issues and clarifications that wold be appreciated.

1. Introduction

- spell-out all acronyms at the first instance.  (e.g., ISNCSCI)

- It's great that the authors introduce the past work on ANN for SCI prognosis and problems with existing studies (lack of comparison to standard approach and small training data)

- It would improve the paper if additional literature search is presented in regard to types of ANN use, accuracy obtained, etc.

- rationale for ANN 

- Aims are clear.

2. Methods

- please include flow chart of inclusion and cohorts (ambulatory vs. non)

- please include more detailed inclusion/exclusion criteria, demographics, age, initial scores at admission, etc.

- was SCIM score measured again after hospitalization and therapy? Was this the main predicted variable for ANN and linear regression models?  Please clarify.

- Since SCIM is the focus of this paper (and results show it too), perhaps expand on what it measures. (And go further into in Discussion why is it such a strong prognostic factor.)

- what was the output layer predicting in the ANN?

- some important details of the ANN training are missing.  Including data split, hyperparameters, loss function, etc.

- It is odd that ANN was ran 10 times? To my understanding, once trained, the output is always the same for the same input?  I am confused about the "poor reliability" statement on line 147.

- Description of linear regression model needs to clearly indicate input and output. 

3. Results

-  for ANN results, RI and NI calculation is not described sufficiently in methods. Provide equations when possible.

-  For Table 1 binary parameters, clarify which was assigned 0 and 1. This will make it easier to understand how the parameters associated with the outcome (for linear regression).

- The same for Table 2.

4. Discussion

- Expand upon different types of ANN (architecture, loss function) if the information is available. Are there only two studies on ANN to predict SCIM?

- line 362:  In light of mortality issue, did your study include any deaths, and did you evaluate if ANN can predict mortality? 

Comments on the Quality of English Language

It's fine but please spell-out all acronyms at the first instance.  (e.g., ISNCSCI)

Round 2

Reviewer 1 Report

Comments and Suggestions for Authors

All of the corrections noted have been properly addressed. Thank you for the appropriate response.